# Counteranion-mediated efficient iodine capture in a hexacationic imidazolium organic cage enabled by multiple non-covalent interactions

Jian Yang[1], Shao-Jun Hu[1], Li-Xuan Cai[1], Li-Peng Zhou[1] & Qing-Fu Sun [1,2] ✉

Developing efficient adsorbents to capture radioactive iodine produced from nuclear wastes is highly desired. Here we report the facial synthesis of a hexacationic imidazolium organic cage and its iodine adsorption properties. Crucial role of counteranions has been disclosed for iodine capture with this cage, where distinct iodine capture behaviors were observed when different counteranions were used. Mechanistic investigations, especially with the X-ray crystallographic analysis of the iodine-loaded sample, allowed the direct visualization of the iodine binding modes at the molecular level. A network of multiple non-covalent interactions including hydrogen bonds, halogen bonds, anion···π interactions, electrostatic interaction between polyiodides and the hexacationic skeleton of the cage are found responsible for the observed high iodine capture performance. Our results may provide an alternative strategy to design efficient iodine adsorbents.

Nuclear power, regarded as a clean and efficient energy source, has attracted considerable attention to meet the increasing worldwide energy demand and the goal of carbon neutralization. However, radioactive wastes, especially the volatile and metabolic-related radionuclides iodine ($^{129}$I, $^{131}$I), existing mainly as elemental iodine together with minor organic iodides and metal iodides from nuclear fuel reprocessing can cause severe environmental and health concerns, which is particularly challenging to tackle and can easily spread through the atmosphere[1–3]. It is therefore of great urgency to develop reliable and efficient adsorbents to capture and store iodine properly. Accordingly, various types of adsorbents, including zeolites, aerogels, metal-organic frameworks (MOFs), covalent organic frameworks (COFs) and porous organic polymers (POPs) have been studied for iodine capture[4–19]. Recently, discrete molecular cages or macrocycles have emerged as promising alternatives as iodine adsorbents[20–27]. For example, nitrogen-rich bipyridine-based organic cages are reported with high iodine capture capacity[24,25]. Pillarene-derived nonporous adaptive crystals (NACs) reported by Huang's group also exhibited

good iodine capture properties[23]. Very recently, Bardelang and Martinez et al. reported a molecular host–guest crystal for energy-efficient iodine uptake[28]. Despite significant progress being made, easily accessible and reliable adsorbents with high affinity and capacity toward iodine is still a challenging task. Besides, investigation of the iodine adsorption mechanism at the molecular level is crucial to addressing the structure-property relationships, which, however, is less studied[22,23,27,29].

To date, most of the known iodine adsorbents are charge neutral. These neutral adsorbents are suggested to capture iodine through weak charge-transfer interactions between adsorbents and electron-deficient iodine[11,14,25]. One strategy for enhancing the affinity and adsorption capacity toward iodine is to introduce cationic units into the adsorbents, where the positively charged adsorbents can strongly adsorb the dynamically formed polyiodide anions through electrostatic interactions[30,31]. For example, Han et al. reported an ionic covalent organic framework via ammonium functionalization with exceptionally high iodine capture capacity[31]. To be mentioned, most

[1]State Key Laboratory of Structural Chemistry Fujian Institute of Research on the Structure of Matter Chinese Academy of Sciences, Fuzhou 350002, PR China.
[2]University of Chinese Academy of Sciences, Beijing 100049, PR China. ✉e-mail: qfsun@fjirsm.ac.cn

reported adsorbents for iodine are shown to capture iodine via solo interaction of either charge-transfer interaction, hydrogen bonding interaction or other supramolecular interactions, which largely limits the affinity and capacity for iodine[23,27,32].

Imidazolium-containing supramolecular architectures, mainly macrocycles, represent a unique class of fascinating hosts that find wide applications not only in recognition[33–41], but also as N-heterocyclic carbene precursors[42–46]. However, compared to the well-studied pyridinium-based structures[47–55], 3D imidazolium organic cages synthesized with irreversible imidazolium C-N bond formation via direct $S_N2$ reactions are very rare[56,57], although efficient synthesis of polyimidazolium-containing organic cages has recently been reported by the Han group through a metal-carbene template approach[42,44]. While studies for iodine capture by imidazolium organic cages have never been investigated, we anticipate that a porous cationic imidazolium organic cage with halide counteranions may serve as a promising iodine adsorbent considering its multiple hydrogen/halogen binding sites, enriched aromatic surfaces, and high charge density.

Here, we report the facial one-pot synthesis of a new hexacationic imidazolium organic cage $3\cdot6X$ (X = Cl⁻, Br⁻, I⁻, PF₆⁻) and its counteranion-mediated iodine capture properties. Impressively, $3\cdot6Cl$ exhibits a record iodine vapor uptake capacity of 5.89 g g⁻¹, breaking the porous organic cage's (POC's) record value of 5.64 g g⁻¹ [24]. Meanwhile, $3\cdot6I$ showed an ultrafast iodine adsorption rate in organic solution, and $3\cdot6PF_6$ can be used for iodine removal from aqueous media. X-ray crystallographic study on the iodine-loaded $3\cdot[I_2Br]_5[I_4Br]$ disclosed that a network of multiple non-covalent interactions, including hydrogen bonds, halogen bonds, anion···π interactions, and electrostatic interactions between polyiodides and the positively charged adaptive skeleton of 3 synergistically contribute to the high iodine capture capacity.

## Results

$3\cdot6Br$ is readily synthesized in one pot from 1 (1,3-di(1H-imidazol-1-yl)benzene) and 2 (1,3,5-tris(bromomethyl)benzene) in a ratio of 3:2. Dropwise addition of 2 into a diluted solution of 1 in CH₃CN under a reflux temperature for 2 days gave rise to a white precipitate, which is recrystallized from water to give $3\cdot6Br$ in 42% yield (Fig. 1). The purity and composition of $3\cdot6Br$ were further confirmed by nuclear magnetic resonance spectroscopy (NMR), electro-spray ionization time-of-flight mass spectrometry (ESI-TOF-MS), as well as X-ray crystallography (Fig. 2b, Supplementary Figs. 2–5). Such a high yield of 42% is quite surprising considering the one-pot formation of six irreversible C-N bonds through $S_N2$ reactions in the absence of additional templates.

For investigating the role of counteranions in iodine capture, we prepared corresponding $3\cdot6X$ (X = PF₆⁻, Cl⁻, I⁻) through anion metathesis with different counteranions, considering their various molecular weight and hydrogen/halogen bond accepting ability. They are all characterized by NMR spectroscopy, ESI-TOF-MS and X-ray crystallography (Figs. 2a, c, d and Supplementary Figs. 7–10, 12–15, 17–20). Interestingly, ¹H NMR spectra showed that the proton $H_c$ in $3\cdot6Cl$ exhibited the largest chemical shift value of 10.3 ppm compared to

10.2, 9.9 and 9.6 ppm in $3\cdot6Br$, $3\cdot6I$ and $3\cdot6PF_6$, respectively (Supplementary Figs. 2, 7, 12, 17), which is in agreement with their ability of acting as a hydrogen bond acceptor with an order of Cl⁻ > Br⁻ > I⁻ > PF₆⁻.

The molecular structures of $3\cdot6X$ (X = Cl⁻, Br⁻, I⁻, PF₆⁻) were definitely determined by X-ray crystallographic analysis. Single crystals of $3\cdot6X$ (X = Cl⁻, Br⁻, I⁻, PF₆⁻) were obtained by either slow evaporation or vapor diffusion methods. In the solid state, $3\cdot6Cl$ crystallized in the $P2_1/c$ space group, where the three diimidazolium units arranged in a parallel fashion (Fig. 2a). A supramolecular chain was formed by inserting one diimidazolium into another two from adjacent $3\cdot6Cl$ (Supplementary Figs. 27a, b). The chains were further stacked into a 3D framework through intermolecular π···π and hydrogen bonding interactions (Supplementary Fig. 27c). $3\cdot6Br$ and $3\cdot6I$ possess a similar conformation to that of $3\cdot6Cl$, with different distances between diimidazolium units (Fig. 2b, c and Supplementary Figs. 28–30). A bromide was found residing in the cavity of $3\cdot6Br$, which was suggested to act as an internal template to draw together three lateral diimidazole units through C − H···Br hydrogen bonding interactions (Fig. 2b). Thus, the preorganisation of the three diimidazole units directed by the bromide may explain the high-yield formation of cage $3\cdot6Br$. Different from a flat conformation observed in $3\cdot6X$ (X = Cl⁻, Br⁻, I⁻), $3\cdot6PF_6$ adopted a pseudo-$C_3$ symmetric arrangement with a PF₆⁻ locating inside the cavity (Fig. 2d). Cationic counterpart 3 was separated and further connected by PF₆⁻ to form a compact 3D supramolecular structure through C-H···F hydrogen bonds (Supplementary Fig. 31). It is noteworthy that X-ray crystallographic studies have clearly elucidated conformation-adaptive skeleton of 3 toward different anions, which provides a structural foundation for adaptive iodine uptake.

Before evaluating iodine capture capacities of $3\cdot6X$ (X = Cl⁻, Br⁻, I⁻, PF₆⁻), a series of measurements were implemented to characterize their morphology, porosity and thermal stability. Powder X-ray diffraction (PXRD) analyses revealed that all four desolvated $3\cdot6X$ (X = Cl⁻, Br⁻, I⁻, PF₆⁻) are actually amorphous (Supplementary Figs. 32–35). Nitrogen adsorption experiments indicated mesoporous nature of $3\cdot6X$ (X = Cl⁻, Br⁻, I⁻, PF₆⁻) with relatively low Brunauer-Emmett-Teller (BET) surface areas (Supplementary Figs. 36–39, Supplementary Table 1). Thermogravimetric analyses (TGA) showed that all four cages $3\cdot6X$ (X = Cl⁻, Br⁻, I⁻, PF₆⁻) exhibit high thermal stability (Supplementary Figs. 40–43). The combination of mesoporous feature, inherently cationic nature and multiple binding sites of these cages encouraged us to investigate their iodine adsorption performances.

Placing the activated $3\cdot6Cl$ under an atmosphere of iodine vapor at 75 °C resulted in the color change of $3\cdot6Cl$ from white to black over time, indicative of iodine adsorption by $3\cdot6Cl$ (Supplementary Fig. 21a). The maximum iodine vapor adsorption capacity reached up to 5.89 g g⁻¹ after 24 h, which was calculated through gravimetric measurements performed at different time intervals (Fig. 3a). It is worth highlighting that the iodine vapor uptake capacity of $3\cdot6Cl$ surpasses all reported organic cage adsorbents like BPPOC (5.64 g g⁻¹)[24], OMC3 (3.78 g g⁻¹)[20], Bpy-cage (3.23 g g⁻¹)[25], and CC3 (0.55 g g⁻¹)[22], which is even comparable to most porous adsorbents (Supplementary

**Fig. 1 | Synthesis of 3·6Br.** Synthetic route of 3·6Br.

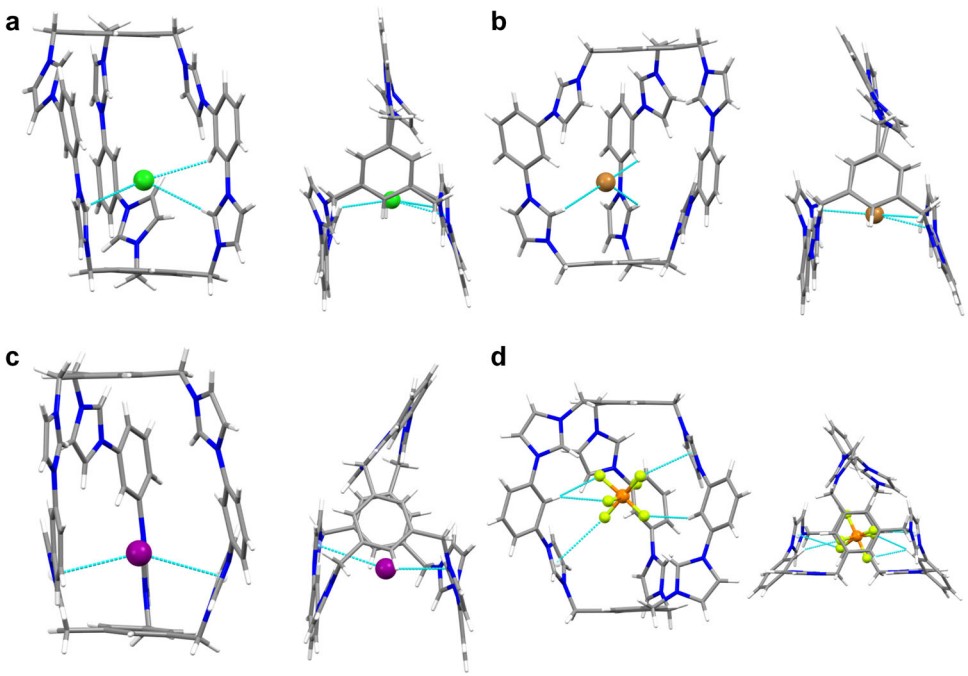

**Fig. 2 | Crystal structures of cages with different counterions.** Side view and top view of **a** **3·**6Cl **b** **3·**6Br **c** **3·**6I and **d** **3·**6PF$_6$. Hydrogen bonds and anion···π interactions are shown with a cyan dashed line. Color code: C, grey; H, white; N, blue; Cl, green; Br, brown; I, purple; P, orange; F, olive.

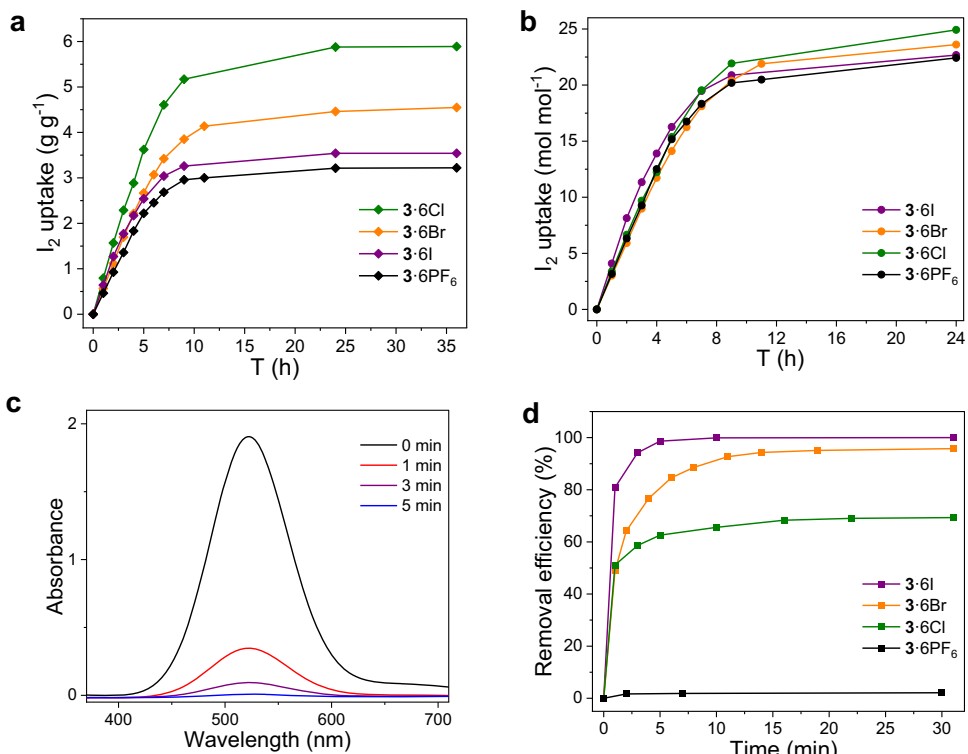

**Fig. 3 | Iodine adsorption experiments.** Time-dependent I$_2$ vapor uptake by **3·**6X (X = Cl$^-$, Br$^-$, I$^-$, PF$_6^-$) in **a** g g$^{-1}$ and **b** mol mol$^{-1}$, respectively. **c** Time-dependent UV/vis absorption spectra of a solution of I$_2$ in n-hexane (2 mM, 3 mL) upon addition of 5 mg **3·**6I. **d** Iodine removal efficiency in solution (2 mM, 3 mL) at various times by 5 mg **3·**6X (X = Cl$^-$, Br$^-$, I$^-$, PF$_6^-$), monitored by change of I$_2$ absorbance at 522 nm.

Table 7)[58,59]. Interestingly, under the same conditions, the iodine vapor uptake capacity is determined to be 4.55 g g$^{-1}$, 3.54 g g$^{-1}$ and 3.22 g g$^{-1}$ for **3·**6Br, **3·**6I and **3·**6PF$_6$, respectively, suggesting the counteranion-dependent iodine adsorption performances (Fig. 3a). For a better comparison of their I$_2$ vapor adsorption capacities and considering the difference of their molecular weights, we recalculated the corresponding data with mol mol$^{-1}$ instead of g g$^{-1}$. As shown in Fig. 3b, 50.1, 48.2, 45.4 and 44.1 iodine atoms are adsorbed by per cage molecule **3·**6Cl, **3·**6Br, **3·**6I and **3·**6PF$_6$, respectively, which exhibited better performances than other cage adsorbents[20,22,23,25].

In addition to iodine vapor, $3\cdot6X$ (X = Cl⁻, Br⁻, I⁻, PF₆⁻) also demonstrated iodine uptake capability from solution. Exposing 5 mg of $3\cdot6X$ (X = Cl⁻, Br⁻, I⁻, PF₆⁻) into a solution of iodine in n-hexane (2 mM, 3 mL) led to color fading, indicative of iodine adsorption (Supplementary Fig. 21b). Unexpectedly, the adsorption process monitored by time-dependent UV/vis spectroscopy indicated that $3\cdot6I$ exhibited the fastest $I_2$ adsorption rate, with the solution of $I_2$ becoming colorless in only 5 min (Fig. 3c and Supplementary Figs. 48–51), though its iodine vapor adsorption capacity is moderately high (Fig. 3a). This phenomenon also happened to $I_2$ vapor adsorption experiments (Fig. 3b). On the contrary, the adsorption of $I_2$ in n-hexane did not finish in the presence of $3\cdot6Cl$ even after 24 h (Supplementary Fig. 48). This could be explained by the nature of polarization-driven halogen⋯halogen contacts, where more polarized iodine⋯iodine interactions tend to be favored compared to halogen bonds involving bromine or chlorine atoms[60]. The $I_2$ adsorption capacity from n-hexane solution by $3\cdot6I$ was eventually determined to be $2.26 \pm 0.02\ g\ g^{-1}$ (Supplementary methods 1.3 Iodine uptake experiments from solution), which was less than that from vapor ($3.54\ g\ g^{-1}$), we infer this might be due to occupation of pores in $3\cdot6I$ by solvent molecules.

It is noteworthy that the iodine adsorption behavior varied depending on the medium and temperature. The iodine uptake rate from solution phase by $3\cdot6X$ (X = Cl⁻, Br⁻, I⁻, PF₆⁻) sticks to the order of polarizabilities of halogen bond acceptors (I⁻ > Br⁻ > Cl⁻ > PF₆⁻) (Fig. 3d). However, the deviation from this order is observed from the vapor phase at high temperature, where $3\cdot6PF_6$ showed a more rapid iodine vapor adsorption rate than $3\cdot6Br$ (Fig. 3b). This may be ascribed to a bigger pore size owned by $3\cdot6PF_6$ than $3\cdot6Br$ (Supplementary Table 1). To be mentioned, the iodine adsorption capacity performed at room temperature with a reduced $I_2$ concentration was lower than those at high temperature (Supplementary Fig. 22a), probably because $I_2$ molecules could not penetrate into inner pores of materials with a reduced iodine concentration. What's more, the subtle differences observed in Fig. 3b were proven to be more pronounced at room temperature with a reduced iodine concentration, and iodine adsorption capacity by per cage molecule was found consistent with the order of polarizabilities of halogen bond acceptors (I⁻ > Br⁻ > Cl⁻ > PF₆⁻) (Supplementary Fig. 22b), which further confirmed the dominant role of halogen bonding in iodine adsorption. We thus infer that the iodine vapor adsorption behavior is governed by multiple factors including polarizabilities of halogen bond acceptors, the pore size of adsorbents and temperature.

Apart from iodine vapor, removal of iodine from aqueous phase is another urgent task, since contaminated water from nuclear reactors containing radioactive iodine poses a long-term threat. Placing 5 mg of water-insoluble $3\cdot6PF_6$ into an aqueous solution[26] of $I_2$/KI (0.4 mM, 3 mL) caused rapid solution color fading from yellow to colorless (Supplementary Fig. 55). Time-dependent UV/vis spectroscopy revealed that the removal efficiency of iodine reached up to > 99% in only one minute in the presence of $3\cdot6PF_6$, which is much faster than most adsorbents for iodine removal from water[25,26,61,62], and $3\cdot6PF_6$ exhibited an adsorption capacity of $1.10 \pm 0.17\ g\ g^{-1}$ from an aqueous solution of $I_2$/KI, indicating its great potential for removing iodine from aqueous phase (Supplementary methods 1.3 Iodine uptake experiments from solution).

We then checked the reusability of our material. Immersing $I_2@3\cdot6Br$ into ethanol resulted in the release of iodine gradually over time with the solution color changing from colorless to dark brown (Supplementary Fig. 23). After iodine desorption, $3\cdot6Br$ was reactivated and used for iodine capture, 86% capacity was reserved after three cycles, indicating excellent recyclability (Supplementary Fig. 24). It is to mention that the adsorbed iodine is hard to remove completely from our cage material due to strong multiple noncovalent interactions (see below), which is also observed for other cage absorbents[21,24].

A series of measurements were carried out to explore the underneath mechanism for the efficient and counteranion-dependent iodine adsorption behaviors of $3\cdot6X$ (X = Cl⁻, Br⁻, I⁻, PF₆⁻), including Fourier transform infrared spectroscopy (FT−IR), NMR, Raman spectroscopy, X-ray photoelectron spectroscopy (XPS), TGA and X-ray crystallography.

The FT−IR spectra of $3\cdot6X$ (X = Cl⁻, Br⁻, I⁻) recorded before and after iodine vapor adsorption showed obvious changes. For example, the C−N stretching vibrations of the imidazole ring at 1122, 1116, and 1115 cm⁻¹ were shifted to 1101, 1103, and 1104 cm⁻¹ for $I_2@3\cdot6Cl$, $I_2@3\cdot6Br$ and $I_2@3\cdot6I$, respectively, suggesting strong interactions between $I_2$ and $3\cdot6X$ (X = Cl⁻, Br⁻, I⁻) (Supplementary Figs. 56–58)[63]. On the contrary, no obvious changes were observed for $3\cdot6PF_6$ after adsorption of iodine, indicative of weak interactions for the imidazole rings on $3\cdot6PF_6$ with iodine (Supplementary Fig. 59). TGA revealed the presence of considerable iodine molecules in the $I_2$-loaded samples (Supplementary Figs. 44–47). It is worth noting that the adsorbed amount of iodine in $3\cdot6X$ (X = Cl⁻, Br⁻) determined by TGA before the decomposition temperature of $3\cdot6X$ is less than that calculated through direct gravimetric measurements of $3\cdot6X$ before and after the $I_2$ uptake. This can be reasonably explained as that some strongly adsorbed $I_2$ especially in the cavity of $3\cdot6X$ through multiple noncovalent interactions escaped accompanied by collapse of the framework of $3\cdot6X$, thus leading to a relatively lower $I_2$ uptake capacity calculated by TGA. Raman spectroscopy and XPS were measured to identify the iodine species after adsorption by $3\cdot6X$ (X = Cl⁻, Br⁻, I⁻, PF₆⁻). After iodine vapor adsorption, two main Raman bands were observed for $I_2@3\cdot6Cl$ at 108 and 148 cm⁻¹, where the former band at 108 cm⁻¹ can be ascribed to symmetric stretching vibration of $I_3^-$, and stretching vibration at 148 cm⁻¹ could be attributed to $[I_2Cl]^-$ (Fig. 4a)[64–66]. As for $I_2@3\cdot6Br$, in addition to the characteristic stretching vibrations at 111 and 165 cm⁻¹ for polyiodide species $I_3^-$ and $I_5^-$ respectively, one additional band at 132 cm⁻¹ was also observed, which can be assigned to the stretching vibration of $[I_2Br]^-$ (Supplementary Fig. 60)[21,64,67]. The participation of halide in the formation of hetropolyhalides was further confirmed by XPS spectra with the observation of the Cl 2p peak shifted from 196.6 to 196.9 eV (Fig. 4b), and the Br 3d peak shifted from 67.2 to 68.1 eV (Supplementary Fig. 63) for iodine-adsorbed samples $I_2@3\cdot6Cl$ and $I_2@3\cdot6Br$, respectively. Moreover, changes in N 1s XPS spectra from imidazolium ring of $3\cdot6X$ (X = Cl⁻, Br⁻, I⁻, PF₆⁻) recorded before and after iodine absorption suggested charge transfer interactions between imidazolium ring and iodine species (Supplementary Figs. 66–69)[24,31].

Raman spectra and I 3d XPS spectra also confirmed the existence of $I_3^-$ and $I_5^-$ in $I_2@3\cdot6I$ and $I_2@3\cdot6PF_6$ (Supplementary Figs. 61–62, 64–65)[21]. ¹H NMR spectra were recorded to monitor the process of the formation of $[I_2X]^-$ (X = Cl⁻, Br⁻, I⁻) through gradual titration of $I_2$ into the solution of $3\cdot6X$ (X = Cl⁻, Br⁻, I⁻, PF₆⁻) in DMSO−$d_6$. Upon addition of $I_2$, structures of $3\cdot6X$ (X = Cl⁻, Br⁻, I⁻) experienced strong chemical perturbations, reflecting on significant chemical shifts for $H_c$ upfielded of 0.90, 0.54 and 0.26 ppm for $3\cdot6Cl$, $3\cdot6Br$ and $3\cdot6I$, respectively (Fig. 4c). As for $3\cdot6PF_6$, no noticeable chemical shifts were observed upon addition of $I_2$ (Fig. 4c), confirming again the lack of strong interactions of $3\cdot6PF_6$ with $I_2$. This is also consistent with the order of hydrogen/halogen bond accepting ability Cl⁻ > Br⁻ > I⁻ > PF₆⁻. It was to note that the ¹H NMR spectra shifted no more when more than 6 equiv of $I_2$ was added, which is in line with six halides present in $3\cdot6X$ (X = Cl⁻, Br⁻, I⁻) (Supplementary Figs. 70–73).

X-ray crystallography is the most powerful method to directly elucidate the specific interactions between different species. Fortunately, we succeeded in obtaining the single crystals of iodine-loaded $3\cdot[I_2Br]_5[I_4Br]$ by slow evaporation of the corresponding solution of $3\cdot6Br$ with $I_2$ in acetonitrile. X-ray crystallographic analysis revealed that $3\cdot[I_2Br]_5[I_4Br]$ crystallizes in the P-1 space group, where one cation 3, together with five $[I_2Br]^-$ and one $[I_4Br]^-$ as counteranions

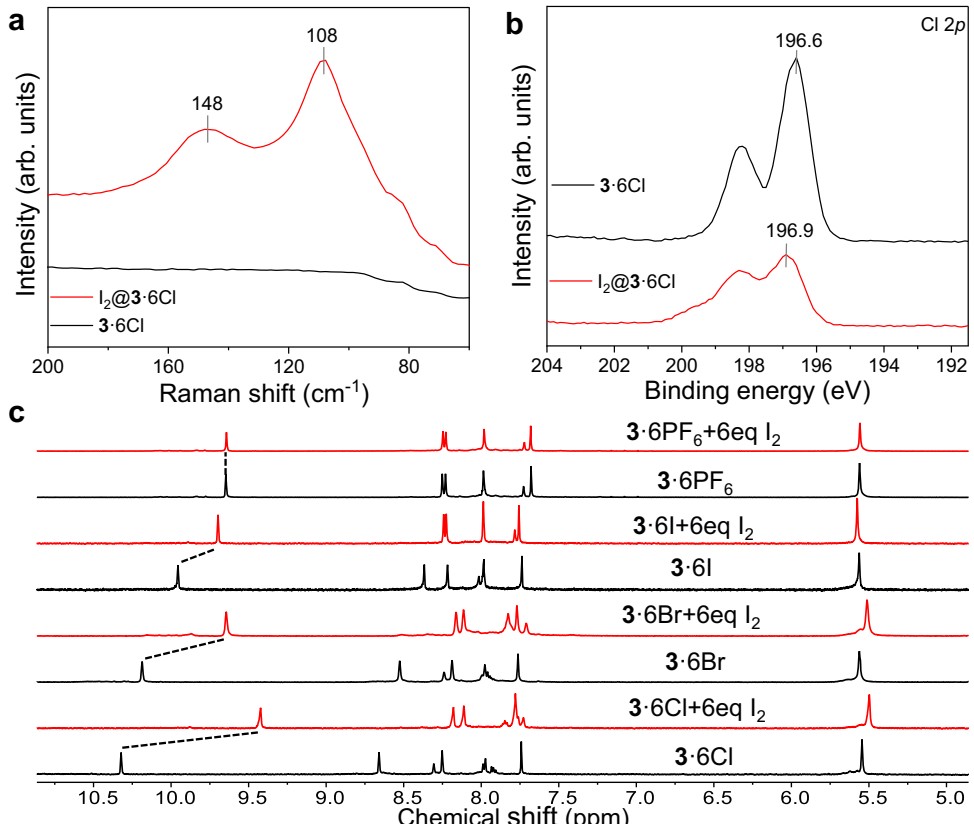

**Fig. 4 | Characterization of the iodine capture. a** Raman spectra of **3**·6Cl before and after $I_2$ adsorption. **b** Cl 2*p* XPS spectra of pristine **3**·6Cl and $I_2$@**3**·6Cl. **c** Partial $^1$H NMR spectra of **3**·6X (X = Cl⁻, Br⁻, I⁻, PF₆⁻) (black line) and corresponding samples with 6 equiv of $I_2$ (red line) in DMSO−$d_6$. Chemical shifts of imidazolium proton $H_c$ induced by addition of $I_2$ are indicated by black dashed lines.

were located in the asymmetric unit (Fig. 5a). The formation of $[I_4Br]^-$ was further evidenced based on mass spectroscopy (Supplementary Figs. 25–26). It is worth noticing that, the possibility of the existence of $[I_5]^-$ could not be excluded from a crystallographic point of view. The Br-I·I angles in $[I_2Br]^-$ range from 175.8° to 178.2°, which is consistent with values in the literature[60,68]. Taking a close look at the structure of **3**·$[I_2Br]_5[I_4Br]$, there is one $[I_2Br]^-$ leaning inside the cavity, anchored by multiple C − H···Br, C − H···I hydrogen bonding and anion···π interactions (Fig. 5c). The adjacent cationic **3** are linked together to form a 1D supramolecular chain in a head-to-tail style through C − H···Br and C − H···I hydrogen bonds (Fig. 5c), which is further extended into a 3D supramolecular framework by a network of short contacts including C − H···Br and C − H···I hydrogen bonds, I − I···I and I − I···Br halogen bonds as well as anion···π interactions. It is noteworthy that after adsorption of $I_2$, the conformation of **3** became less flat compared to the pristine **3**·6Br, and the tight packing of the cages is broken, where the voids are filled with polyiodides.

The above findings allow us to propose that $I_2$ was firstly captured by X to form $[I_2X]^-$ (X = Cl⁻, Br⁻, I⁻) via strong halogen bonding interaction, which is then anchored by multiple hydrogen bonds, halogen bonds, anion···π interactions, as well as strong electrostatic interaction with the highly positively charged **3**. The conformation of adaptive skeleton of **3** was rearranged by the formed polyiodides at high temperature, which prevents tight packing of the cages, and meanwhile creates more accessible voids with smaller halides for diffusion and adsorption of $I_2$, leading to eventually high and different iodine capture performance of **3**·6X (X = Cl⁻, Br⁻, I⁻) (Fig. 3a, b). On the other hand, for **3**·6PF₆, only weak interaction occurred between cylindrical electron surface of the I − I bond and imidazolium C − H protons of **3**[25,69]. Besides, PF₆⁻ sitting inside the cavity of **3** occupied much place that

could be used to store iodine, which is responsible for the relatively low iodine capture capacity by **3**·6PF₆. Moreover, iodine vapor capture experiments conducted at room temperature highlighted the important role of halogen bonding interaction in iodine adsorption (Supplementary Fig. 22b). We infer that the enhanced iodine vapor uptake capacity originates not only from the use of low-molecular-weight and stronger halogen bond accepting halide, but also probably from more voids created by smaller halides between adaptive cage structures.

## Discussion

In summary, we have prepared an easily accessible and conformation-adaptive imidazolium porous organic cage with rich charge density and multiple noncovalent binding sites. Different from most reported charge-neutral adsorbents, our hexacationic imidazolium organic cage exhibited a record iodine vapor adsorption capacity among POCs, due to the combination of its mesoporous nature and multiple non-covalent interactions, including hydrogen bonds, halogen bonds, anion···π interactions, and strong electrostatic interaction. The crucial role of counteranions is revealed, showing that low-molecular-weight and stronger halogen/hydrogen bond accepting halide are beneficial for high iodine adsorption capacity. The present work not only provides a good platform to study iodine adsorption behavior at the molecular level, but also expands the applications of imidazolium organic cages.

## Methods

### Materials and instrumentation

All chemicals and solvents were purchased from Adamas-beta with analytical reagent purity and used without further purification unless otherwise noted. NMR spectra were recorded on a Bruker Biospin

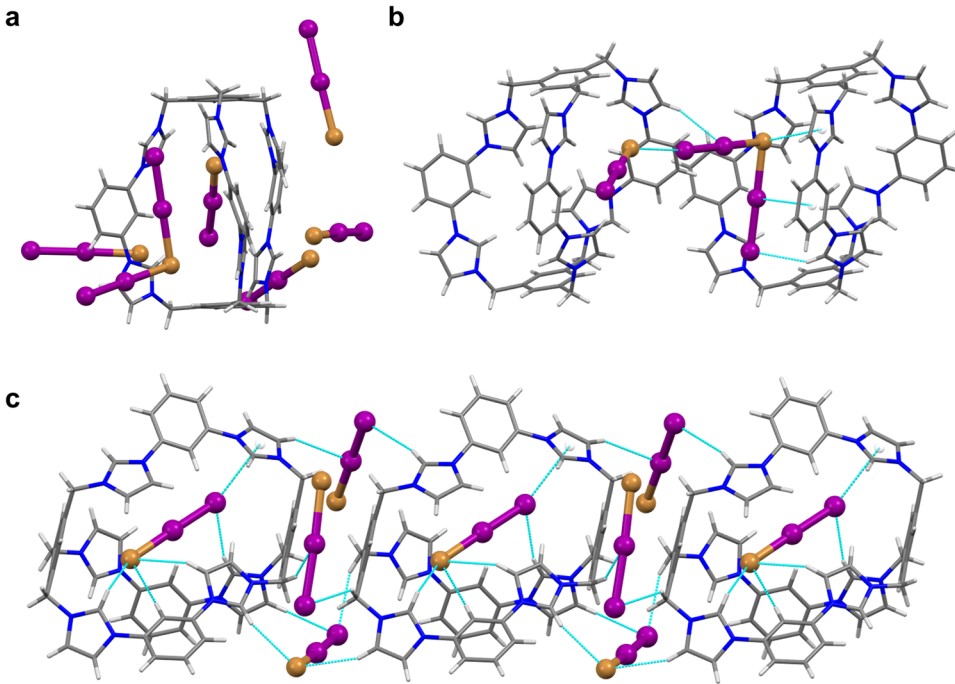

**Fig. 5 | X-ray crystal structure of 3·[I₂Br]₅[I₄Br].** **a** Asymmetric unit containing one cation **3**, one [I₄Br]⁻ and four [I₂Br]⁻ as counteranions. **b** Stabilization of [I₄Br]⁻ by C − H···I hydrogen bonds, I − I···Br halogen bond and anion···π interactions. **c** Side view of one dimensional chain connected by [I₂Br]⁻ anions through multiple C − H···Br and C − H···I hydrogen bonds with one [I₂Br]⁻ inside the cavity. Short contacts are shown by cyan dashed lines. Color code: C, grey; H, white; N, blue; Br, brown; I, purple.

Avance III (400 MHz) spectrometer and JEOL ECZ600S (600 MHz) spectrometer. Chemical shifts are given in ppm with respect to TMS or residual signals of the deuterated solvents used. UV-Vis adsorption spectra were recorded on UV-2700 UV-Visible spectrophotometer from SHIMADZU Corporation. High-resolution ESI-TOF-MS were recorded on an Impact II UHR-TOF mass spectrometry from Bruker, with sodium formate and sodium trifluoroacetate as the internal standard. Data analysis was conducted with the Bruker Data Analysis software (Version 4.3) and simulations were performed with the Bruker Isotope Pattern software. Powder X-ray diffraction measurements were carried out with a Miniflex 600 diffractometer equipped with Cu Kα radiation (λ = 1.5418 Å). Thermogravimetric analysis (TGA) was investigated on a NETZSCH STA449F3 unit under N2 atmosphere at a heating rate of 10 K min-1. Nitrogen adsorption/ desorption isotherms were measured with an ASAP 2020 based on the Brunauer-Emmett-Teller (BET) method. Fourier transform infrared (FT-IR) spectra were recorded on the Bruker VERTEX70 system by mixing samples into KBr to prepare a compressed tablet sample. X-ray photoelectron spectro-scopy (XPS) was conducted on a Thermo Fisher ESCALAB 250Xi by using an Al Kα (λ = 8 Å, hν = 1486.6 eV) X-ray source without any etching. The Raman spectra were recorded on a LabRAM HR Evolution spectrometer equipped with a 785 nm laser.

### Syntheses of cage compounds 3·6X (X = Cl⁻, Br⁻, I⁻, PF₆⁻)

To 100 mL acetonitrile solution of **1** (50 mg, 0.24 mmol, 1.5 equiv) in a 250 mL flame-dried round bottom flask was added **2** (57 mg, 0.16 mmol, 1 equiv) in 100 mL acetonitrile dropwise. The reaction mixture was stirred at reflux temperature for 2 d to give rise to pure **3·6Br** as a white precipitate from the reaction solution, which was recrystallized from water, collected and dried (45 mg, 42% yield). **3·6X** (X = Cl⁻, I⁻, PF₆⁻) were readily obtained through anion metathesis with different counteranions. See Synthesis and characterization section in the Supplementary Information for more details on the synthesis of all of the compounds described in this paper.

## Data availability

X-ray crystallographic data for the structures reported in the article have been deposited at the Cambridge Crystallographic Data Centre, under deposition numbers CCDC 2234458 (**3·6Cl**), 2234459 (**3·6Br**), 2234460 (**3·6I**), 2234461 (**3·6PF₆**), 2234462 (**3·[I₂Br]₅[I₄Br]**). Copies of the data can be obtained free of charge via https://www.ccdc.cam.ac.uk/ structures/. The dataset is also provided as Supplementary Data 1 with this paper. All relevant data are available in this paper and its Supplementary Information, and additional data are available from the corresponding author upon request. Source data are provided with this paper.

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

## Acknowledgements

This work was supported by the National Key Research and Development Program of China (2021YFA1500400 (Q.-F.S.)), National Natural Science Foundation of China (Grant Nos. 22001252 (J.Y.), 21825107 (Q.-F.S.), 22171264 (Q.-F.S.), 22171262 (L.-X.C.)) and Science Foundation of Fujian Province (Grant Nos. 2022J05093 (J.Y.) and 2021J02016 (Q.-F.S.)).

## Author contributions

Q.-F.S and J.Y. conceived and designed this project. J.Y. carried out the synthesis, characterization, and iodine adsorption study. S.-J.H. and L.-X.C. assisted with crystal structure refinement and analysis. L.-P.Z. assisted with mass spectroscopy measurement structural determination. J.Y. and Q.-F.S. wrote the manuscript. All the authors discussed the results and commented on the manuscript.

## Competing interests

The authors declare no competing interests.
