## [Peer Review File · Nature Communications]

Counteranion-Mediated Efficient Iodine Capture in a Hexacationic Imidazolium Organic Cage Enabled by Multiple Non-covalent InteractionsREVIEWER COMMENTS

Reviewer #1 (Remarks to the Author):

In this paper, the authors reported the facial synthesis of a hexacationic imidazolium organic cage 3·6X (X = Cl⁻, Br⁻, I⁻, PF₆⁻) with a quite high yield in the absence of additional templates. Impressively, 3·6X exhibits excellent iodine adsorption performance as demonstrated by the record-high iodine vapor uptake capacity of 3·6Cl among the thus far reported porous organic cages and fast adsorption kinetics in organic and aqueous solution for 3·6I and 3·6PF₆⁻, respectively. Moreover, the iodine adsorption mechanism was also investigated. In particular, the direct visualization of the binding modes at the molecular was revealed according to X-ray crystallographic analysis. The novelty of this work is sufficient and can be published in Nat. Commun. after the minor revision.

(1) Please give the full name of NMR, ESI-TOF-MS, and other abbreviations mentioned for the first time.

(2) In order to fully reflect the changes in C-N bonds before and after iodine absorption, the N1s XPS spectra of 3·6X before and after iodine absorption are suggested to be provided.

(3) In pages S7 and S10, "36+·6Cl-" and "36+·6I-" are suggested to change to "3·6Cl" and "3·6I", respectively.

Reviewer #2 (Remarks to the Author):

In this article Sun et al describe a hexacationic imidazolium porous organic cage for iodine capture. They synthesis a new cage, with various counterion, and obtained nice X-ray structures of these compounds, even after reaction with iodine. They demonstrate that their systems, after activation, are amorphous and can capture iodine in vapor or iodine in solution (water or hexane). The vapor uptake capacity is high. For cage with Cl⁻, Br⁻ and I⁻ as counterion, it is a very simple and well-known reaction between these anions and I₂ that forms I₂-X (Figures 4 and 5). As a consequence, there is an expected role of the counterion on iodine uptake: the rate of the reaction is different depending on the anion. Thus, this article might be suitable for publication in Nature Communications after the following points have been addressed:

Major comments:

- The reaction between iodine and halide anion is a very classical and well-known reaction. To highlight the interest of their structures, the authors have to compare the ability of iodine uptake by their cage structure with those of other simple halide imidazolium salts or even halide ammonium salts. Such salts may be able to induce the same chemical reaction with iodine. These could help to highlight the interest and originality of their approach and their specific structure.
- The cage with PF₆ as counterion seems efficient for iodine vapor or iodine in water uptake, but not for iodine presents in a hexane solution. What is the explanation? Moreover I₂-X can't be formed with this counterion (supported by the IR, RAMAN and NMR spectroscopies), thus what is the mechanism of iodine uptake with this compound?

Minor comments:

- The NMR spectra in the SI are too small.
- The capture of iodine takes place at rather high temperature, what happen at room temperature?

Reviewer #3 (Remarks to the Author):

In this study, the authors synthesized a hexacationic imidazolium organic cage, 3-6X (X = Cl-, Br-, I-, PF₆-) with relative ease, and for the first time revealed the significant role of its counteranions in iodine capture, both in the vapor and solution phases. Remarkably, 3-6Cl demonstrates a record-breaking iodine vapor uptake of 5.89 g g⁻¹, while 3-6I shows an extraordinarily swift iodine adsorption rate in organic solution. Additionally, the water-insoluble 3-6PF₆ can be effectively utilized for iodine removal from aqueous environments. Furthermore, the authors disclosed the intrinsic capture mechanism through an X-ray crystallographic study, suggesting that multiple non-covalent interactions collectively contribute to the high iodine uptake. The manuscript is well-organized, and clearly illustrates the progression of thought. The intriguing results regarding the role of counteranions in different mediums could provide valuable insights for the design of efficient ionic adsorbents in a range of scenarios. However, there are several issues that

need to be addressed before this research can be considered for publication.

1. In the introductory section, the authors mention that radioactive iodine primarily manifests as elemental iodine, organic iodides, and metal iodides. Could the authors specify which metal iodides they are referring to? Please provide further clarification on this matter.

2. In Figure 3b, the authors attribute the variance in the iodine adsorption rate across the four samples to a combined effect of the polarizabilities of halogen bond acceptors and the pore size. However, could the authors further elucidate the structure-property relationship between the iodine adsorption capacity and the adsorbents?

3. The authors conducted the iodine vapor adsorption experiment under a high temperature of 75 °C in a static system, resulting in a very high iodine concentration. I postulate that the subtle differences observed in Figure 3b might be due to this excessively high iodine concentration. Therefore, could the authors possibly conduct the experiment with a reduced iodine concentration to ascertain if the differences among the four adsorbents would be more pronounced? Dynamic adsorption measurement at a low I₂ partial pressure would be another interesting test to unravel the different impacts of various ions on I₂ adsorption.

4. In Figure 4a, the two curves appear to be reversed, and the same issue is observed in Figure S54. Furthermore, there seems to be an error in the labeling of the I₅⁻ peak. I would urge the authors to thoroughly review and correct these mistakes.

5. In the solution phase, the observed iodine uptake rate aligns with the order of polarizabilities of the halogen bond acceptors (I⁻ > Br⁻ > Cl⁻ > PF₆⁻). However, Figure 4c suggests that 3·6Cl induced the largest chemical shift after the introduction of I₂ into its solution in DMSO. Aren't these two findings contradictory? Could the authors provide an explanation or clarification for this?

6. The authors note that the adsorbed amount, as determined by TGA is less than the amount calculated through direct gravimetric measurement. However, my calculations derived from the TGA results align closely with those measured by the gravimetric method. The authors are suggested to reevaluate this conclusion.

7. Recent publications relevant to I₂ capture using porous materials should be acknowledged to better represent the current state of research in the field. For instance, the authors could consider citing "Nature Communications, 13, Article number: 2878

(2022)" and "Chemical Engineering Journal 468, 143525 (2023)". This inclusion would provide a more comprehensive review of the existing literature and advancements.

Point-by-Point Response to Reviewers' Comments

Reviewer #1 (Remarks to the Author):

In this paper, the authors reported the facial synthesis of a hexacationic imidazolium organic cage $3\cdot 6X$ ($X = \text{Cl}^-$, Br^- , Γ , PF_6^-) with a quite high yield in the absence of additional templates. Impressively, $3\cdot 6X$ exhibits excellent iodine adsorption performance as demonstrated by the record-high iodine vapor uptake capacity of $3\cdot 6\text{Cl}$ among the thus far reported porous organic cages and fast adsorption kinetics in organic and aqueous solution for $3\cdot 6\text{I}$ and $3\cdot 6\text{PF}_6$, respectively. Moreover, the iodine adsorption mechanism was also investigated. In particular, the direct visualization of the binding modes at the molecular was revealed according to X-ray crystallographic analysis. The novelty of this work is sufficient and can be published in Nat. Commun. after the minor revision.

Our response: we are grateful to this reviewer for positive and constructive comments about our work. Accordingly, we have carefully addressed the questions posed by this reviewer as listed below.

(1) Please give the full name of NMR, ESI-TOF-MS, and other abbreviations mentioned for the first time.

Our response: we thank this reviewer for pointing out this. And the full names of NMR, ESI-TOF-MS, and other abbreviations mentioned for the first time have been given in the revised manuscript.

Accordingly, the following change has been made in the revised manuscript: NMR was changed to nuclear magnetic resonance spectroscopy (NMR).

(2) In order to fully reflect the changes in C-N bonds before and after iodine adsorption, the N1s XPS spectra of $3\cdot 6X$ before and after iodine adsorption are suggested to be provided.

Our response: we appreciate this valuable suggestion from this reviewer. And accordingly we recorded the N 1s XPS spectra of $3\cdot 6X$ before and after iodine adsorption. Before iodine uptake, $3\cdot 6\text{Cl}$ displayed two peaks at 401.5 and 399.2 eV,

respectively, which can be assigned to imidazolium ring of the cage. After iodine adsorption, the peak at 399.2 eV shifted to 399.8 eV, indicating charge transfer interactions between imidazolium ring and iodine species. Similar changes were also observed for cages with other counterions $\mathbf{3}\cdot\mathbf{6X}$ ($X = \text{Br}^-$, I^- , PF_6^-) (Figure R1).

Accordingly, the following discussion has been added to the revised manuscript: “Moreover, changes in N_{1s} XPS spectra from imidazolium ring of $\mathbf{3}\cdot\mathbf{6X}$ ($X = \text{Cl}^-$, Br^- , I^- , PF_6^-) recorded before and after iodine absorption suggested charge transfer interactions between imidazolium ring and iodine species.” Besides, N_{1s} XPS spectra of $\mathbf{3}\cdot\mathbf{6X}$ before and after iodine adsorption were also provided in the revised SI.

Figure R1. N_{1s} XPS spectra of $\mathbf{3}\cdot\mathbf{6X}$ before and after I_2 adsorption.

(3) In pages S7 and S10, “ $\mathbf{3}^{6+}\cdot\mathbf{6Cl}^-$ ” and “ $\mathbf{3}^{6+}\cdot\mathbf{6I}^-$ ” are suggested to change to “ $\mathbf{3}\cdot\mathbf{6Cl}^-$ ” and “ $\mathbf{3}\cdot\mathbf{6I}^-$ ”, respectively.

Our response: we thank this reviewer for pointing out this mistake. Corresponding changes have been made as suggested in the revised supplementary information.

Accordingly, the following changes have been made in pages S8 and S10: “ $3^{6+} \cdot 6\text{Cl}^-$ ” and “ $3^{6+} \cdot 6\text{I}^-$ ” were changed to “ $3 \cdot 6\text{Cl}^-$ ” and “ $3 \cdot 6\text{I}^-$ ”, respectively, in the revised supplementary information.

Reviewer #2 (Remarks to the Author):

In this article Sun et al describe a hexacationic imidazolium porous organic cage for iodine capture. They synthesis a new cage, with various counterion, and obtained nice X-ray structures of these compounds, even after reaction with iodine. They demonstrate that their systems, after activation, are amorphous and can capture iodine in vapor or iodine in solution (water or hexane). The vapor uptake capacity is high. For cage with Cl^- , Br^- and I^- as counterion, it is a very simple and well-known reaction between these anions and I_2 that forms $\text{I}_2\text{-X}$ (Figures 4 and 5). As a consequence, there is an expected role of the counterion on iodine uptake: the rate of the reaction is different depending on the anion. Thus, this article might be suitable for publication in Nature Communications after the following points have been addressed:

Our response: we are grateful to this reviewer for positive and constructive comments about our work. Accordingly, we have carefully addressed the questions posed by this reviewer as listed below.

Major comments:

- The reaction between iodine and halide anion is a very classical and well-known reaction. To highlight the interest of their structures, the authors have to compare the ability of iodine uptake by their cage structure with those of other simple halide imidazolium salts or even halide ammonium salts. Such salts may be able to induce the same chemical reaction with iodine. These could help to highlight the interest and originality of their approach and their specific structure.

Our response: we appreciate this valuable suggestion from this reviewer. As suggested, we used simple ammonium halide salts TBAX (TBA = tetrabutylammonium, $\text{X} = \text{Cl}^-$, Br^- , I^-) as adsorbents for iodine adsorption experiments, showing unexpectedly high iodine adsorption capacities with a value of 4.75, 4.26, and 3.48 g g^{-1} for TBACl, TBABr and TBAI, respectively (Figure R2a), which is comparable to our cage

materials. The order of iodine adsorption capacity was consistent with our results, where low-molecular-weight halides contributed to a bigger adsorption capacity (TBACl > TBABr > TBAI). Interestingly, when we transformed unit g g^{-1} into mol mol^{-1} , almost the same amount of I_2 was adsorbed by per TBAX with a value of 5.2, 5.4 and 5.1 for TBACl, TBABr and TBAI, respectively, indicating that the difference of iodine adsorption capacity in g g^{-1} just came from the variation of molecular weights of halide in TBAX. In contrast, each of our cage molecule adsorbed 25, 24 and 22.7 I_2 for **3**·6Cl, **3**·6Br and **3**·6I, respectively. Significant differences in these values can be attributed to adaptive cage structure, where cages with smaller halides created more voids for capturing more I_2 , and this difference is gradually amplified from **3**·6I, **3**·6Br to **3**·6Cl, revealing the structural advantages of our cages. It is noteworthy that, after iodine adsorption, TBAX became a liquid (Figure R2b), and I_2 seemed to “dissolve” in this liquid, thus resulting in a high iodine adsorption capacity. Low molecular weights of TBAX also contribute to this high performance. And the strategy of using adsorbents with halides is also proven to be efficient in this case. However, we think this is a different system from our material, and thus it is not appropriate to make a direct comparison between these two systems. Anyway, we have to admit that the proposal posed by this reviewer inspired us deeply, and relevant research about this kind of compounds as iodine adsorbents will be conducted.

As this is a different system from ours, therefore we do not add this discussion to the revised manuscript.

Figure R2. (a) Iodine adsorption by TBAX ($X = \text{Cl}^-$, Br^- , I^-) at 75 °C, and (b) photograph of TBAI after I_2 adsorption.

- The cage with PF_6^- as counterion seems efficient for iodine vapor or iodine in water uptake, but not for iodine presents in a hexane solution. What is the explanation? Moreover, $\text{I}_2\text{-X}$ can't be formed with this counterion (supported by the IR, RAMAN and NMR spectroscopies), thus what is the mechanism of iodine uptake with this compound?

Our response: we thank this reviewer for the constructive questions. Actually, in the water phase, in order to increase the solubility of iodine, we used I_2/KI as iodine source with the following dynamic equilibrium $\text{I}_2 + \text{I}^- \rightleftharpoons \text{I}_3^-$. And the cage with PF_6^- as counterion is suggested to adsorb I_3^- in water through electrostatic interactions and hydrogen bonding interactions between protons of imidazolium units in the cage and I_3^- . However, in hexane solution, the iodine exists as neutral elemental iodine I_2 , and the interactions between charge neutral I_2 and cationic cage are supposed to be charge transfer interactions, which is quite weak, resulting in less efficiency for I_2 uptake in hexane by the cage with PF_6^- . As for the situation of iodine vapor uptake by $3\cdot6\text{PF}_6$ at 75 °C in a static system with a high iodine concentration, we suppose that under such conditions, some I_3^- and I_5^- species formed as evidenced from XPS and Raman spectroscopies which are attracted by the cationic cage moiety through electrostatic interactions and hydrogen bonding interactions. Besides, concentrated I_2 vapor diffused

into pores of $3\cdot 6\text{PF}_6$, and as discussed in the main text that weak interactions between cylindrical electron surface of the I–I bond and imidazolium C–H protons of the cage could be responsible for the I_2 capture behavior with this compound (*J. Am. Chem. Soc.* **2022**, *144*, 113–117.), and $\text{I}\cdots\pi$ interactions between I_2 and aromatic rings of the cage might also participate in the I_2 capture process (*J. Am. Chem. Soc.* **2021**, *143*, 2325–2330.).

Minor comments:

- The NMR spectra in the SI are too small.

Our response: we thank this reviewer for pointing out this. And the NMR spectra in the SI have been revised.

Accordingly, NMR spectra for Supplementary Figs. 2-4, 7-9, 12-14, 17-19, 68-71 in supplementary information have been reproduced.

- The capture of iodine takes place at rather high temperature, what happen at room temperature?

Our response: we appreciate this valuable suggestion from this reviewer. And accordingly, we performed iodine capture experiment at room temperature. As shown in Figure R3a, the order of iodine adsorption capacity at room temperature is consistent with that at 75 °C ($3\cdot 6\text{Cl} > 3\cdot 6\text{Br} > 3\cdot 6\text{I} > 3\cdot 6\text{PF}_6$), and the capacity is lower than that at high temperature, probably because I_2 molecules could not penetrate into inner pores of materials with a reduced iodine concentration. Moreover, it took longer time for saturated adsorption of iodine. As for iodine capture capacity shown in Figure 3b, at 75 °C in a static system with a high iodine concentration, $3\cdot 6\text{Cl}$ exhibited the largest iodine capture capacity by per cage molecule followed by $3\cdot 6\text{Br}$, $3\cdot 6\text{I}$ and $3\cdot 6\text{PF}_6$, respectively. However, this order was reversed for $3\cdot 6\text{X}$ ($\text{X} = \text{Cl}^-$, Br^- , I^-) when iodine capture experiments were conducted at room temperature with a reduced I_2 concentration (Figure R3b). We postulate that under high temperature with a concentrated I_2 atmosphere, adaptive cages underwent conformation rearrangements upon iodine adsorption, where cages with smaller counterions created more accessible voids for diffusion and adsorption of I_2 , leading to the resulting differences in iodine

adsorption performance in Figure 3b. Whereas under milder conditions (room temperature, reduced I₂ concentration), halogen bonding interaction is supposed to play a dominant role in iodine adsorption, where cages with good halogen bonding acceptors as counterions tend to capture more I₂ (I⁻ > Br⁻ > Cl⁻).

Accordingly, we added the following discussion to the revised manuscript: “To be mentioned, the iodine adsorption capacity performed at room temperature with a reduced I₂ concentration was lower than those at high temperature, probably because I₂ molecules could not penetrate into inner pores of materials with a reduced iodine concentration. What’s more, the subtle differences observed in Fig. 3b were proven to be more pronounced at room temperature with a reduced iodine concentration, and iodine adsorption capacity by per cage molecule was found consistent with the order of halogen bond accepting ability (I⁻ > Br⁻ > Cl⁻ > PF₆⁻), which further confirmed the dominant role of halogen bonding in iodine adsorption.”

Figure R3. (a, b) Time-dependent I₂ vapor uptake by 3·6X (X = Cl⁻, Br⁻, I⁻, PF₆⁻) in g g⁻¹ and mol mol⁻¹ at room temperature, respectively.

Reviewer #3 (Remarks to the Author):

In this study, the authors synthesized a hexacationic imidazolium organic cage, 3·6X (X = Cl⁻, Br⁻, I⁻, PF₆⁻) with relative ease, and for the first time revealed the significant role of its counteranions in iodine capture, both in the vapor and solution phases. Remarkably, 3·6Cl demonstrates a record-breaking iodine vapor uptake of 5.89 g g⁻¹, while 3·6I shows an extraordinarily swift iodine adsorption rate in organic solution. Additionally, the water-insoluble 3·6PF₆ can be effectively utilized for iodine removal

from aqueous environments. Furthermore, the authors disclosed the intrinsic capture mechanism through an X-ray crystallographic study, suggesting that multiple non-covalent interactions collectively contribute to the high iodine uptake. The manuscript is well-organized, and clearly illustrates the progression of thought. The intriguing results regarding the role of counteranions in different mediums could provide valuable insights for the design of efficient ionic adsorbents in a range of scenarios. However, there are several issues that need to be addressed before this research can be considered for publication.

Our response: we are grateful to this reviewer for positive and constructive comments about our work. Accordingly, we have carefully addressed the questions posed by this reviewer as listed below.

1. In the introductory section, the authors mention that radioactive iodine primarily manifests as elemental iodine, organic iodides, and metal iodides. Could the authors specify which metal iodides they are referring to? Please provide further clarification on this matter.

Our response: we thank this reviewer for pointing out this. As stated in the introductory section, radioactive iodine exists mainly as elemental iodine together with minor organic iodides and metal iodides from nuclear fuel reprocessing. According to the literature, between 0.8 and 6% of the iodine resides in the gap region between fuel and cladding and between fuel pellets, typically as CsI, and some of the iodine remains in the dissolver as insoluble colloidal solids, principally as PdI₂ and AgI (*J. Nucl. Mater.* **2016**, *470*, 307–326).

And for clarification, we have added this literature as reference 3 in the revised manuscript.

2. In Figure 3b, the authors attribute the variance in the iodine adsorption rate across the four samples to a combined effect of the polarizabilities of halogen bond acceptors and the pore size. However, could the authors further elucidate the structure-property relationship between the iodine adsorption capacity and the adsorbents?

Our response: we thank this reviewer for pointing out this. As for iodine capture

capacity shown in Figure 3b, at 75 °C in a static system with a high iodine concentration, **3·6Cl** exhibited the largest iodine capture capacity by per cage molecule followed by **3·6Br**, **3·6I** and **3·6PF₆**, respectively. We postulate that under such conditions, adaptive cages underwent conformation rearrangements upon iodine adsorption, where cages with smaller counterions created more accessible voids for diffusion and adsorption of I₂, leading to the resulting differences in iodine adsorption performance in Figure 3b. However, this order is reversed for **3·6X** (X = Cl⁻, Br⁻, I⁻) when iodine capture experiments were conducted at room temperature with a reduced I₂ concentration (Figure R3). We suppose that under such milder conditions, halogen bonding interaction plays a dominant role in iodine adsorption capacity, where cages with good halogen bonding acceptors as counterions tend to capture more I₂ (I⁻ > Br⁻ > Cl⁻). And accordingly, we revised the following discussion in the revised manuscript: “The conformation of adaptive skeleton of **3** was rearranged by the formed polyiodides at high temperature, which prevents tight packing of the cages, and meanwhile creates more accessible voids with smaller halides for diffusion and adsorption of I₂, leading to eventually high and different iodine capture performance of **3·6X** (X = Cl⁻, Br⁻, I⁻).” “Moreover, iodine vapor capture experiments conducted at room temperature highlighted the important role of halogen bonding interaction in iodine adsorption.”

Figure R3. (a, b) Time-dependent I₂ vapor uptake by **3·6X** (X = Cl⁻, Br⁻, I⁻, PF₆⁻) in g g⁻¹ and mol mol⁻¹ at room temperature, respectively.

3. The authors conducted the iodine vapor adsorption experiment under a high temperature of 75 °C in a static system, resulting in a very high iodine concentration. I

postulate that the subtle differences observed in Figure 3b might be due to this excessively high iodine concentration. Therefore, could the authors possibly conduct the experiment with a reduced iodine concentration to ascertain if the differences among the four adsorbents would be more pronounced? Dynamic adsorption measurement at a low I₂ partial pressure would be another interesting test to unravel the different impacts of various ions on I₂ adsorption.

Our response: we appreciate the valuable suggestion from this reviewer. As suggested, we conducted the iodine adsorption experiment at room temperature with a reduced iodine vapor concentration. As shown in Figure R3a, iodine adsorption capacity at room temperature is lower than those at high temperature, probably because I₂ molecules could not penetrate into inner pores of materials with a reduced iodine concentration. And as evidenced in the main text that the iodine uptake rate is consistent with the polarizabilities of halogen bonding acceptors (I⁻ > Br⁻ > Cl⁻ > PF₆⁻), the subtle differences observed in Figure 3b are proven to be more pronounced at room temperature with a reduced iodine concentration (Figure R3b).

Accordingly, the following discussion has been added in the revised manuscript: “To be mentioned, the iodine adsorption capacity performed at room temperature with a reduced I₂ concentration was lower than those at high temperature, probably because I₂ molecules could not penetrate into inner pores of materials with a reduced iodine concentration. What’s more, the subtle differences observed in Fig. 3b were proven to be more pronounced at room temperature with a reduced iodine concentration, and iodine adsorption capacity by per cage molecule was found consistent with the order of halogen bond accepting ability (I⁻ > Br⁻ > Cl⁻ > PF₆⁻), which further confirmed the dominant role of halogen bonding in iodine adsorption.”

We agree with this reviewer that dynamic adsorption measurement at a low I₂ partial pressure would be another interesting measurement, however, we do not have the equipment for I₂ dynamic adsorption for the moment, and this would be another project in our laboratory. Thank you for your understanding.

Figure R3. (a, b) Time-dependent I_2 vapor uptake by $3\cdot 6X$ ($X = Cl^-$, Br^- , I^- , PF_6^-) in $g\ g^{-1}$ and $mol\ mol^{-1}$ at room temperature, respectively.

4. In Figure 4a, the two curves appear to be reversed, and the same issue is observed in Figure S54. Furthermore, there seems to be an error in the labeling of the I_5^- peak. I would urge the authors to thoroughly review and correct these mistakes.

Our response: we thank this reviewer for pointing out this mistake. And we have thoroughly reviewed the manuscript and SI again.

Accordingly, the errors in labeling in Figure 4a and Figure S54 (now Supplementary Fig. 59) have been corrected.

5. In the solution phase, the observed iodine uptake rate aligns with the order of polarizabilities of the halogen bond acceptors ($I^- > Br^- > Cl^- > PF_6^-$). However, Figure 4c suggests that $3\cdot 6Cl$ induced the largest chemical shift after the introduction of I_2 into its solution in DMSO. Aren't these two findings contradictory? Could the authors provide an explanation or clarification for this?

Our response: we thank this reviewer for pointing out this. In the hexane solution, the iodine uptake rate sticks to the polarizabilities of the halogen bond acceptors ($I^- > Br^- > Cl^- > PF_6^-$), which has been reasonably explained by the nature of polarization-driven halogen...halogen interactions. In Figure 4c, upon addition of I_2 into the DMSO solution of cages with different counterions, $3\cdot 6Cl$ suffered from the largest chemical shift. These two results don't contradict each other. As mentioned above, halogen...halogen interactions, i.e. halogen bonds tend to be favored between more polarized halogens ($I^- > Br^- > Cl^-$), (*Angew. Chemie - Int. Ed.* **2013**, 52, 13444–13448).

Specifically, iodide is the largest, most polarizable halide, which means more opportunity to collide with I₂ to form I₃⁻ through halogen bonds, leading to the fastest iodine uptake rate in hexane solution despite being the least electronegative.

However, chemical shifts are the result of the change of electronic density. And prior to I₂ titration, more electronegative Cl⁻ induced the largest chemical shift among **3**·6X (X = Cl⁻, Br⁻, I⁻, PF₆⁻). Upon addition of I₂, the reactions of Cl⁻, Br⁻, I⁻ with I₂ give rise to trihalide I₂Cl⁻, I₂Br⁻ and I₃⁻, respectively, and the difference of electronegativity among these three anions is largely diminished, which is reflected in similar chemical shifts induced by I₂Cl⁻, I₂Br⁻ and I₃⁻ (Figure 4c, red line). Therefore, the difference of electronegativity between Cl⁻ and I₂Cl⁻ is the largest, thus resulting in the largest chemical shift before and after I₂ addition.

And in order to avoiding misunderstanding, we deleted the following sentence in the revised manuscript: “where the formed [I₂Cl]⁻ induced the largest chemical shift.”

6. The authors note that the adsorbed amount, as determined by TGA is less than the amount calculated through direct gravimetric measurement. However, my calculations derived from the TGA results align closely with those measured by the gravimetric method. The authors are suggested to reevaluate this conclusion.

Our response: we thank this reviewer for pointing out this. Actually, for I₂-adsorbed samples I₂@**3**·6Cl and I₂@**3**·6Br, the amount of iodine species was determined to be 80% and 78% by TGA, respectively, which is less than those obtained through direct gravimetric measurement (85% for I₂@**3**·6Cl and 82% for I₂@**3**·6Br). As for I₂@**3**·6I and I₂@**3**·6PF₆, the adsorbed amount of iodine determined by TGA aligns well with those calculated through direct gravimetric measurement.

Accordingly, for precision, we deleted “I⁻” from the statement in the main text “It is worth noting that the adsorbed amount of iodine in **3**·6X (X = Cl⁻, Br⁻) determined by TGA before the decomposition temperature of **3**·6X is less than that caculated through direct gravimetric measurements of **3**·6X before and after the I₂ uptake.” Please see the revised manuscript.

7. Recent publications relevant to I₂ capture using porous materials should be

acknowledged to better represent the current state of research in the field. For instance, the authors could consider citing "Nature Communications, 13, Article number: 2878 (2022)" and "Chemical Engineering Journal 468, 143525 (2023)". This inclusion would provide a more comprehensive review of the existing literature and advancements.

Our response: we thank this reviewer for pointing out this.

Accordingly, the two suggested articles about I₂ capture have been cited in the revised manuscript as reference 18 and 19.

REVIEWERS' COMMENTS

Reviewer #1 (Remarks to the Author):

Since all the issues concerned have been clarified in the present version of manuscript, I suggest acceptance of this paper for publication in Nature Communications.

Reviewer #2 (Remarks to the Author):

The authors have clearly answer to my previous comments. They have performed additional experiments and the science described here is robust and very interesting. My only concern is about the adsorption abilities of the TBAX salts that are comparable to their cages. We could wonder about the interest of a cage system for I₂ capture when simple TBA salts present similar properties. It could be also highlighted that their systems have also low molecular weight per halide, lower than TBAX salts (Six halide as counterion per cage). Thus, I think that the sentence "Low molecular weights of TBAX also contribute to this high performance" is not a relevant here. Nevertheless, I agree that the behavior of The TBAX salts seems different from the cages because the adsorption capacity change with the nature halide a little bit more with the cages (around 10%) than with the TBA salts (around 5%) and because TBAX become liquid after I₂ adsorption. In agreement with the authors, this latter point suggests that I₂ may then dissolved in the liquid, making a direct comparison with the cage difficult. Thus, the behavior of these two system strongly differ, and the fact the TBAX become liquid after I₂ adsorption, whereas the cages stay solid under the same conditions, also highlight the interest and relevance of their system. Therefore, I think that this article may be now suitable for publication in Nature Communications.

Reviewer #3 (Remarks to the Author):

The revised manuscript has addressed the questions and concerns raised during the first round of review. I recommend it to be accepted for publication.

Point-by-Point Response to Reviewers' Comments

Reviewer #1 (Remarks to the Author):

Since all the issues concerned have been clarified in the present version of manuscript, I suggest acceptance of this paper for publication in Nature Communications.

Our response: We are grateful to this reviewer for recommending this paper for publication in Nature Communications.

Reviewer #2 (Remarks to the Author):

The authors have clearly answer to my previous comments. They have performed additional experiments and the science described here is robust and very interesting. My only concern is about the adsorption abilities of the TBAX salts that are comparable to their cages. We could wonder about the interest of a cage system for I₂ capture when simple TBA salts present similar properties. It could be also highlighted that their systems have also low molecular weight per halide, lower than TBAX salts (Six halide as counterion per cage). Thus, I think that the sentence “Low molecular weights of TBAX also contribute to this high performance” is not a relevant here. Nevertheless, I agree that the behavior of The TBAX salts seems different from the cages because the adsorption capacity change with the nature halide a little bit more with the cages (around 10%) than with the TBA salts (around 5%) and because TBAX become liquid after I₂ adsorption. In agreement with the authors, this latter point suggests that I₂ may then dissolved in the liquid, making a direct comparison with the cage difficult. Thus, the behavior of these two system strongly differ, and the fact the TBAX become liquid after I₂ adsorption, whereas the cages stay solid under the same conditions, also highlight the interest and relevance of their system.

Our response: We appreciate this reviewer's professional and constructive comments. The control experiments suggested by this reviewer strongly helped to highlight the interest and originality of our approach and our specific structure. However, as replied in the first round of review, TBAX and our cage materials demonstrate strikingly different iodine adsorption behaviors especially theirs states after iodine adsorption

(liquid state versus solid state), making a direct comparison between these two kinds of compounds difficult. Thus we decided not to add this discussion to the revised manuscript, considering the integrity, logic and structure of the manuscript.

Therefore, I think that this article may be now suitable for publication in Nature Communications.

Our response: We are grateful to this reviewer for recommending this paper for publication in Nature Communications.

Reviewer #3 (Remarks to the Author):

The revised manuscript has addressed the questions and concerns raised during the first round of review. I recommend it to be accepted for publication.

Our response: We are grateful to this reviewer for recommending this paper for publication in Nature Communications.